# Recent Metal Nanotheranostics for Cancer Diagnosis and Therapy: A Review

**DOI:** 10.3390/diagnostics13050833

**Published:** 2023-02-22

**Authors:** Amir Khorasani, Daryoush Shahbazi-Gahrouei, Arash Safari

**Affiliations:** 1Department of Medical Physics, School of Medicine, Isfahan University of Medical Sciences, Isfahan 81746-73461, Iran; 2Department of Radiology, Ionizing and Non-Ionizing Radiation Protection Research Center (INIRPRC), School of Paramedical Sciences, Shiraz University of Medical Sciences, Shiraz 71439-14693, Iran

**Keywords:** metal nanoparticles, nanotheranostics, medical imaging, radiotherapy, cancer

## Abstract

In recent years, there has been an increasing interest in using nanoparticles in the medical sciences. Today, metal nanoparticles have many applications in medicine for tumor visualization, drug delivery, and early diagnosis, with different modalities such as X-ray imaging, computed tomography (CT), magnetic resonance imaging (MRI), positron emission tomography (PET), etc., and treatment with radiation. This paper reviews recent findings of recent metal nanotheranostics in medical imaging and therapy. The study offers some critical insights into using different types of metal nanoparticles in medicine for cancer detection and treatment purposes. The data of this review study were gathered from multiple scientific citation websites such as Google Scholar, PubMed, Scopus, and Web of Science up through the end of January 2023. In the literature, many metal nanoparticles are used for medical applications. However, due to their high abundance, low price, and high performance for visualization and treatment, nanoparticles such as gold, bismuth, tungsten, tantalum, ytterbium, gadolinium, silver, iron, platinum, and lead have been investigated in this review study. This paper has highlighted the importance of gold, gadolinium, and iron-based metal nanoparticles in different forms for tumor visualization and treatment in medical applications due to their ease of functionalization, low toxicity, and superior biocompatibility.

## 1. Introduction

Nanoparticles in medicine have applications in developing novel therapeutic and diagnostic modalities for cancer treatment and detection [1]. In recent years, there has been growing interest in using nanoparticles in medicine, especially in medical imaging as a contrast agent, in radiation therapy as carriers for drug and gene delivery, and as a radio-sensitizer [2,3,4].

Radiation therapy plays an essential role in cancer treatment. A primary concern of radiation therapy is to maximize tumor damage and minimize healthy tissue damage. Recent developments in the field of nanoparticles have led to a renewed interest in using metal nanoparticles in cancer treatment as radio-sensitizers [4]. It has been suggested [5] that for extended periods of circulate time of NPs as a detoxification device through the blood, we can use NPs coated with red blood cell (RBC) membranes.

Medical imaging is an increasingly important area in the medical sciences. In recent years, there has been an increasing interest in developing contrast agents for different imaging modalities such as computed tomography (CT), magnetic resonance imaging (MRI), radiography, positron emission tomography (PET), single photon emission computed tomography (SPECT), etc. One of the most significant current discussions in medical imaging is metal nanoparticle usage as a contrast agent for image contrast enhancement [6]. Many attempts have been made [7,8,9,10,11,12] to use an NP near-infrared (NIR) fluorescent dye as a contrast agent for molecular imaging. It has been suggested [7] that using near-infrared (NIR) fluorescent dye, IR-26, as a contrast agent preferentially accumulates in the mitochondria of acute myeloid leukemia (AML) cells, and this seems to be an innovative approach for AML targeting, detection, and therapy.

Many metal and metallic elements can form nanostructures that we can use in cancer detection and treatment. Metal nanoparticles are a major area of interest in nanoparticle usage in medicine due to their unique physical and chemical properties, such as magnetic, optical, thermal, catalytic, and electrical properties compared with other NPs [4,6]. There is a considerable amount of literature on using different metal nanoparticles such as gold, bismuth, tungsten, tantalum, ytterbium, gadolinium, silver, iron, platinum, lead, etc., with different forms such as a solid, porous, antibody, folic-acid functionalized, core-shell, and coated, and these were used in various imaging modalities such as magnetic resonance imaging (MRI), computed tomography (CT), radiology, nuclear medicine, and radiation therapy with a different mode for cancer detection and treatment [7,8,9]. Although extensive research has been carried out on developing different metal NPs in the medical sciences for tumor detection and treatment [13,14,15], the need for a single study exists that compares updated research results. This paper will review the new research conducted on using different metal NPs in medical imaging and radiation therapy for cancer detection and treatment. Until recently, some research has been carried out on different metal NPs in medical sciences for cancer detection and treatment, but no single study exists which has reviewed new updates on using metal NPs for cancer detection and treatment. This review study was performed on the recent literature sourced from scientific citation websites such as Google Scholar, PubMed, Scopus, and Web of Science up through the end of January 2023. All relevant works published on the mentioned scientific citation websites were investigated. This review aims to describe new findings in the literature on metal nanoparticles used in medical imaging and radiation therapy. The overall structure of the study consists of two sections including a review of the research on the recent nanotheranostics used in radiation therapy and medical imaging and also a summary of new findings and perspectives.

## 2. Nanotheranostics

Several nanotheranostics are used in medical imaging and radiation therapy for tumor detection and treatment. Their applications in medicine are presented individually.

### 2.1. Gold-Based Nanoparticles

Many studies have used gold nanoparticles (AuNPs) for therapeutic applications in cancer treatments [16] (Figure 1). High atomic number, relatively strong photoelectric absorption coefficient, good renal clearance, and biocompatibility are the features that make AuNPs a good, promising choice for use in radiotherapy [17,18]. Studies have often investigated the radiation sensitization and synergistic effects of AuNPs alone or in combination with other materials or treatment modalities. One pioneering in-vivo study was conducted by Herold et al. [19] in 2000 when it was first reported that gold microspheres could produce radiation dose enhancement against tumors using kilovoltage X-rays. Tudda et al. reported that 15 nm AuNPs could produce biologically effective dose enhancement in rotational radiotherapy of breast cancer using kilovoltage X-rays [20]. Luan et al. [21] described an improvement in the radio-therapeutic efficiency in the treatment of esophageal tumor-bearing mice following the delivery of AuNPs to tumors. Particle type, radiation parameters, and cell type are the main factors that affect the radio-sensitization efficiency of AuNPs. The sensitizing or synergistic effects of AuNPs in radiation therapy have been investigated using in vitro studies with X-rays, γ-rays, electron beams, and high-energy charged protons/carbon ions. In a study, the results showed that AuNPs could produce sensitization and synergistic effects in radiotherapy using different types of radiation [22,23,24,25]. In a major advance in 2022, Mzwd et al. [26] used a green technique for nanoparticle synthesis. They formed and used stable AuNPs in gum arabic (GA) solution via laser ablation technique as a CT contrast agent. They claim that the image CT numbers increased with the concentration of GA-AuNPs. It has been suggested that the GA-AuNPs can be used as a CT contrast agent. Moreover, in another study that set out to determine the effect of glucose-modified dendrimer-entrapped gold nanoparticles (Au DENPs) labeled with radionuclide ^68^Ga for positron emission tomography (PET)/ CT dual-mode imaging, Li et al. [27] found that ^68^Ga labeled with 2-amino-2-deoxy-D-glucose (DG) DG-Au DENPs can be used for PET/CT imaging and immunotherapy of different tumor types. In this line [2], the researcher investigated the multi-modality imaging and photothermal effect of gold-doped upconverting nanoparticles (UCNPs). Zhang et al. [28] pointed out that due to photothermal stability, low cytotoxicity, and high biocompatibility, Au-UCNPs-DSPE-PEG_2k_ may be utilized as MRI and CT contrast agents for both in vivo and in vitro, and may also be used for photothermal treatment.

Furthermore, the size, shape, surface functionalization, concentration, and intracellular distribution of AuNPs can influence their effect on radiation [29,30,31,32]. In 2020, in our department, in a published paper [33], authors synthesized applied alginate-coated iron oxide-gold core-shell nanoparticles (Fe_3_O_4_@Au/Alg NPs) for synergistic photo-thermo-radiotherapy. They found that the presence of gold nanoparticles in the synthesized nanocomposite significantly improves the photothermal efficiency and dose-enhancement of X-rays to a great extent. They further proposed that ionizing radiation exposure was cell cycle phase-dependent for cellular uptake of Fe_3_O_4_@Au/Alg NPs. Additionally, their results demonstrated that the radiation-induced delay of cell division and association of cells in the radio-sensitive cell cycle phase (like G2/M) could enhance the radio-sensitization effect of Fe_3_O_4_@Au/Alg NPs in tumor cells. In [34], the authors investigated NIR-II photo/chemodynamic therapy properties of gold nanobipyramids and copper sulfide in a core/shell architecture (AuNBP@CuS) for cancer treatment and achieved positive results. In another study in our department, AS1411 aptamer-targeted ultrasmall gold nanoclusters (Apt–GNCs) were synthesized, and they showed the ability of Apt–GNCs for radiation enhancement [35]. Apt–GNCs significantly enhanced radiotherapy efficacy, as mean tumor volume decreased by about 39%, and a nine-day increase in mice survival was observed. Both GNCs and Apt–GNCs were biocompatible [35]. Kitayama et al. [36] also reported that when combined with low-dose X-ray radiation therapy, the novel stealth radiation sensitizer based on Au-embedded, molecularly imprinted polymer nanogels (Au MIP-NGs) could inhibit in-vivo tumor growth. Recently, Baijal et al. [37] evaluated the therapeutic and imaging effect of PEGylated gold NPs (Au-PEG-NPs) and silver NPs (Ag-PEG-NPs) as radio-sensitizers and CT contrast agents in the oral cancer KB cell line. They claimed that Au-PEG-NPs exhibit better radio-sensitizer and contrast agent performance than Ag-PEG-NPs. It has been suggested that the Au-PEG-NPs could be used as a radiosensitizer and CT contrast agent candidate for oral cancer treatment and detection, which shows it is a theranostics agent. For instance, the application of gold-based nanoparticles as theranostics is shown in Figure 1 [38]. Surveys such as that conducted by Hu et al. [39] have shown that carbon-based, dot-capped gold nanoparticles (AuNPs/CDs) are promising photosensitizers for photodynamic therapy applications. The most striking result to emerge from the literature reviews in gold-based NPs is that the gold-based NPs, due to their special features, are good clinical candidates for use in cancer detection and treatment.

**Figure 1 diagnostics-13-00833-f001:**
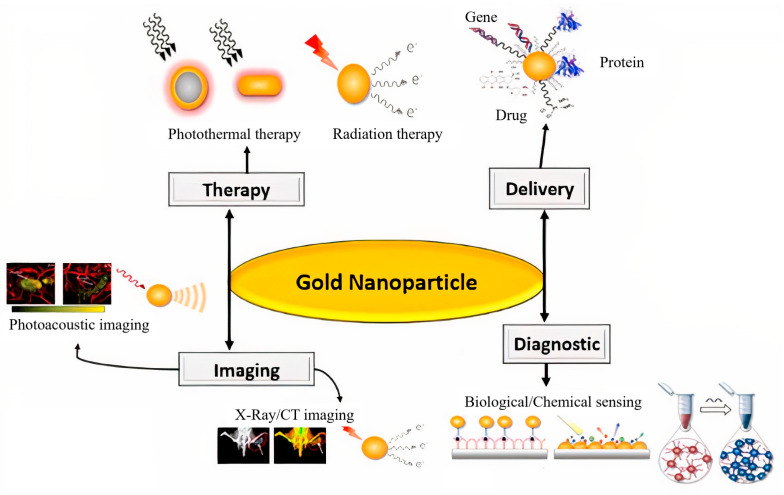
Application of gold-Based nanoparticles in medicinal [38]. “Reprinted with permission from Ref. [38]. 2020”. “Copyright and Licensing” are available via the following link: https://www.frontiersin.org/articles/10.3389/fchem.2020.00376/full.

### 2.2. Gadolinium-Based Nanoparticles 

Gadolinium (Gd, rare earth (lanthanide) metal, Z = 64)-based nanoparticles have been used as multifunctional theranostic (diagnostic and therapeutic) agents in MRI-guided radiotherapy. Hence, Gd chelates have been applied for a more precise, accurate, and enhanced dose delivery in radiotherapy [40,41]. A study in 1996 [42] was conducted on the radio-sensitizing effect of Gd (III) texaphyrin (Gd-tex^2+^). Results of this study showed that Gd-tex^2+^ was established to be an efficient radiation sensitizer in in-vitro and in-vivo experiments carried out with HT29 cells and a murine mammary carcinoma model, respectively. Motexafin gadolinium (MGd), a metallotexaphyrin, is a compound of gadolinium and an expanded porphyrin that can enhance the cytotoxic effects of radiation through several mechanisms relying on the additional generation of reactive oxygen species (ROS) that catalyze the oxidation of intracellular-reducing metabolites and interference with repair mechanisms of radiation-induced damage, which lead to increased cell death. Motexafin gadolinium showed great promise for multifunctional theranostic applications, especially for glioma treatment, and so far, two-phase clinical studies combined with standard radiation treatment have been established [43]. The research results indicated that, in addition to exploiting GdNPs as a positive MR imaging T_1_ contrast agent, they had been identified as valued theranostic sensitizers for radiation therapy [44,45]. Being a toxic lanthanide heavy metal, free GdNPs are not used clinically; instead, it is used as an organic chelating agent compound. The GdNP “Activation and Guidance of Irradiation by X-ray,” or AGuIX, is a polysiloxane nanoparticle with chelated gadolinium that exhibits no toxicity in preclinical and early-stage clinical studies in humans at medically used concentrations and is eliminated rapidly via the kidneys [46,47,48]. The radiation dose enhancement and synergistic effects of GdNPs have also been proven in combination with other ionizing radiation types, such as γ-rays, X-rays, and charged particles [49,50,51]. At kilovoltage X- and γ-ray energies, interactions with high-Z GdNPs produce photoelectrons and numerous Auger electrons, which short-ranged electrons may improve the killing effects of radiation in a highly localized region, on the order of a few cell diameters or less [52,53]. With the increase in energy in megavoltage energies (MV), the physical mechanisms of radiation sensitization become less important and give way to biological mechanisms such as immune responses, oxidative stress, DNA damage, and repair responses [54,55]. The investigation into the combination of cell therapy and nanotechnology found that gadolinium-neutron capture therapy (Gd-NCT) can be used for glioblastoma multiforme (GBM) treatment [56]. More recent evidence [57] highlights that the Au@DTDTPA(Gd) NPs, in combination with conventional external X-ray irradiation, may be used as a radio-sensitizer for GBM treatment.

Today, due to the T_1_ and T_2_ shortening relaxation time effect, gadolinium plays a crucial role in MR imaging. In a study that aimed to synthesize and characterize FeGdPt NPs, Chou et al. [58] found that FeGdPt NPs have feasible applications in dual-modal MRI (T_1_/T_2_) and CT imaging. More recent evidence [59] shows that coated porous silicon NPs (pSiNPs) with gadolinium ions (Gd^3+^) can be used as an MRI contrast agent. They point out that pSiNPs-Gd showed high drug encapsulation efficacy and T_1_-weighted MR image contrast agent performance. Bennettet al. [41] investigated the uptake of GdNPs in patients with pancreatic cancer for MR-guided radiation therapy. It has been suggested [60] that glucosamine (GlcN) conjugated with polyacrylic acid (PAA)-coated ultrasmall gadolinium oxide NPs (UGONs), GlcN-PAA-UGONs, have higher contrast enhancement in T_1_-weighted MR images, and GlcN-PAA-UGONs seem to be an excellent T_1_-weighted MRI contrast agent. Eriksson et al. [61] used a survey to improve T_1_-weighted image contrast at the lower dose of Gd ions in cerium oxide NPs as a T_1_-weighted MRI contrast agent and achieved positive results. It has been demonstrated that a high T_1_ relaxivity of Gd oxide (Gd_2_O_3_) conjugated with mesoporous silica NPs (Gd_2_O_3_@MSNPs) in comparison with Gd diethylene triamine pentaacetate (Gd-DTPA) results in in-vivo use of Gd_2_O_3_@MSNPs as a T_1_-weighted MR image contrast agent [62]. In an investigation into the clinical use of ultrasmall GdNPs (Gd@PEG NPs), Wang et al. [63] found that monodispersible ultrasmall GdNPs can be used for diagnosis of kidney dysfunction through the in-vivo T_1_-weighted MR imaging. These findings further support the idea of using gadolinium-based NPs for cancer detection and imaging, especially as an MRI contrast agent for functional and molecular imaging with T_1_ and T_2_ mapping.

### 2.3. Iron-Based Nanoparticles

Iron-based NPs that include inorganic paramagnetic iron oxide (or magnetite) nanoparticles or superparamagnetic iron oxide nanoparticles (SPIONs) have been investigated as theranostic magnetic nanoparticles and are ideal agents for theranostic applications, especially cancer treatment due to their excellent properties, such as facile synthesis, biocompatibility, and biodegradability [1]. Iron NPs are useful as excellent MRI agents, photothermal therapy (PTT), photodynamic therapy (PDT), magnetic hyperthermia, radiation therapy, and chemo/biotherapeutics presented in varied investigations [64]. Iron NPs could be used as radio-sensitizers/enhancers. Although radio-sensitization is usually proposed for high Z-metals, and the atomic number of iron (Fe, Z = 26) is relatively low, IONs are primarily used in combination with low-linear energy transfer (LET) keV and MV X-rays. Iron oxide nanoparticles increased ROS production in cancer cells when combined with radiation therapy, compared to the treatment with radiation therapy alone [65]. The efficiency of the radio-sensitization potential of silica-coated iron oxide magnetic nanoparticles (SIONPs) when exposed to an X-ray beam was studied in MCF-7 cells. MCF-7 cells tend to show increased radio-sensitization enhancement; meanwhile, with 0.5 Gy dose, dose enhancement factor (DEF) values of cells treated with 5 and 10 μg/mL of SIONPs were 1.21 and 1.32, respectively. Results demonstrated that SIONPs potentially improve the radio-sensitivity of breast cancer [66]. Guerra et al. [67] studied the radio-sensitization effects produced by gold and dextran-coated superparamagnetic iron oxide nanoparticles (SPION-DX) in M059J and U87 human glioblastoma cell lines irradiated by 6 MV photons beam. For U87 cells, SPION-DX nanoparticles with a core diameter of 21.1 nm showed a maximum sensitization enhancement ratio (SER10%) = 1.61 in the group exposed to 50 µg/mL of nanoparticles. For the radio-sensitive M059J cells, sensitization assisted by both types of nanoparticles was much less efficient. Furthermore, they found that sensitization mechanisms occurring through GNPs mostly follow the promotion of lethal complex damage, but SPION-DX repairable damage dominates. Other studies also reported the enhancement of radio-sensitization and synergistic effects on tumor cells in vitro and in vivo using X-rays accompanying with SPIONs and IONs [68,69,70]. Recently, in several studies, iron-based nanoparticle-mediated radio-sensitization was observed in combination with low-energy X-ray and monoenergetic γ-ray radiation [71,72,73]. Most of them reported that IONs enhanced the efficacy of X-ray energies above Fe K-edge more significantly than conventional broadband high-energy X-rays. Although the FDA has approved several ION formulations, specific unwanted toxicity issues reported in many studies that could be overcome by functionalization and surface modification with various coverage and ligands would be helpful to improve their circulation time, clearance, and evasion by reticuloendothelial system, as well as improving tissue targeting, biocompatibility, and stability [74,75,76,77,78]. It has now been demonstrated that [79] the doxorubicin (DOX)-loaded liposomal iron oxide NPs (IONP) (Lipo-IONP/DOX) might serve as a safe and effective agent for combined chemo/photothermal cancer therapy. Currently, iron oxide NPs are the ideal agents for cancer theranostics.

A large and growing body of literature has investigated the impact of iron nanoparticles in medical imaging, especially in MRI. Fe_3_O_4_/Ag_3_VO_4_/Au three-component coated with Caerophyllum macropodum extract modified with oleic acid has been identified as a contrast agent for MRI and CT imaging [80]. Recent evidence suggests that super-ferromagnetic iron oxide nanoparticle chains (SFMIOs) can improve MR image resolution and signal-to-noise ratio (SNR) [81]. In their analysis of iron nanoparticles (Fe NPs) for medical imaging, Dash et al. [82] identified that Fe NPs have high stability against oxidation and exhibit a much stronger shortening of the T_2_ relaxation time in MR imaging. In a study that set out to determine the usage of hypoxia-triggered, self-assembling, ultrasmall iron oxide (UIO) NPs for tumor hypoxia map detection, Zhou et al. [83] found that UIONPs amplify T_2_-weighted signals of ROI in the MRI images and could be considered as a potential nanoprobe candidate for hypoxia imaging of tumors. Detailed examination of paramagnetic ferric iron (III) ion-chelated poly (lactic-co-glycolic) acid NPs properties for T_1_-weighted MR imaging by Marasini et al. [84] showed that this contrast agent has three times the relevant magnetic field relaxivity compared to the commercial agent gadopentetate dimeglumine, Magnevist^®^. In a major study, researchers [85] identified characteristics such as a remarkable biosafety profile, prolonged blood circulation time upon proper surface modification, and renal clearance capacity of ultrasmall superparamagnetic iron oxide (USPIO) NPs as a T_1_/T_2_ weighted MRI contrast agent. It has now been suggested that [86] the polyethylene glycol-coated ultrasmall superparamagnetic iron oxide nanoparticle-coupled sialyl Lewis X (USPIO-PEG-sLe^x^) NPs can reduce the T_2_^*^ value of nasopharyngeal carcinoma (NPC) tumors. They claim that USPIO-PEG-sLe^x^ can be used as a nanotheranostic platform. Lee et al. conducted several discussions and analyses of NPs’ biodistribution and pharmacokinetics with a non-radioactive method [87]. They suggested that iron oxide NPs coated with chitosan and polyethylene glycol can be used for assessing the pharmacokinetics properties of NPs.

In recent years, there has been an increasing amount of literature on using magnetic particle imaging (MPI) for tumor detection [88]. MPI is a novel noninvasive molecular imaging technology that images the bio-distribution of SPIONs [88]. MPI is a favorable tool for cancer diagnosis, as it offers the advantages of zero background signal, zero signal reduction with increasing tissue depth, quantitative linearity, and high sensitivity. Additionally, there is no need for ionizing radiation. Surveys such as that conducted by Chandrasekharan et al. [89] have shown that, in addition to imaging, SPIONs can also be used for tumor ablation with hyperthermia as theranostic NPs. It has conclusively been shown [90] that MPI can be used for cell tracking and detection with high sensitivity and specificity. The previous study [91] has reported that clinical Resovist^®^, superparamagnetic iron oxide NPs, can be used for MPI imaging clinically. In [92,93], the authors investigated the clinical performance of Resovist^®^ for liver imaging with MRI. It has been suggested [94,95] that NanoTherm^®^, an iron oxide NPs agent, can be heated by an externally applied alternating magnetic field for the clinical treatment of solid tumors. In this study, iron-based NPs were found to be a perfect candidate for cancer detection and treatment due to iron’s physical and chemical properties.

### 2.4. Tungsten-Based Nanoparticles

Tungsten (W, Z = 74) can produce photoelectrons, Compton electrons, scattered photons, high-energies characteristic X-rays, positron and negative electron pairs, and Auger electrons under high-energy irradiation that result in direct and indirect interactions (free radicals) with tumor cells [25,96,97,98]. Qin et al. [99] revealed that tungsten nanoparticles could be used for photothermal therapy (PTT) and RT combination treatment. Chen et al. [100] designed a novel theranostics nanoplatform (Au NPs/UCNPs/WO_3_@C) comprising tungsten trioxide (WO_3_) that loaded gold nanoparticles (Au NPs) and up-conversion nanoparticles (UCNPs). The nanosystem exhibited superior oxygen-generation effects and doxorubicin loading capacity, thus serving as an efficient radio-sensitizer for radio-chemo anticancer therapy. Niknam et al. discussed tungsten disulfide (WS2)-based nanomaterial as a PTT agent. In combination with X-ray irradiation, the nanocomposite could catalyze the high expression of H_2_O_2_ to produce cell membrane disruption, mitochondrial dysfunction, reactive oxygen species (ROS) production, and oxidative stress. The results showed that local RT/PTT could efficiently inhibit tumor metastasis, ablate local tumors, and prevent the recurrence of tumors. At the same time, the nanocomposite could also induce high temperatures under near-infrared irradiation to enhance RT results [97]. A large and growing body of literature [101,102,103,104] has investigated tungsten nanoparticles’ radiation protection and shielding effect in medical imaging. Surveys such as that conducted by Wu et al. [105] have shown that ultrasmall metal cores and metal-oxide shell nanoparticles, such as CoFe-WO_x_ (CoWO_4_-Fe_2_WO_6_-WO_3_), can be used as theragnostic nanoprobes for visible/infrared/MRI/CT imaging and photothermal/photodynamic and magnetothermal/magneto-dynamic therapies. The first study of two-dimensional (2D) PEGylated WO_2.9_ (a substoichiometric form of WO_3_) nanosheets for multimodal imaging was reported by Zhang et al. [106] in 2022. In another major study, Chen et al. [100] found that Au NPs and up-conversion NPs (UCNPs) loaded with tungsten trioxide (WO_3_) produce novel theragnostic NPs, Au NPs/UCNPs/WO_3_@C, which improved PA imaging performance. The research of Li et al. [107] showed that it is possible to use thermo-responsive polyethylene glycol-coated tungsten-doped vanadium dioxide (W-VO_2_@PEG) NPs as nanoprobes for depth PA imaging.

### 2.5. Platinum-Based Nanoparticles

Platinum nanoparticles (PtNPs) are relatively new agents that have been extensively used as part of anticancer drug formulation (cisplatin, carboplatin, and oxaliplatin, etc.) in chemotherapy and chemoradiotherapy [108]. Considering the effective antioxidant property and anti-tyrosinase activity of PtNPs, developing these nanoparticles as anticancer agents can be one of the most valuable approaches for clinical use [109,110]. In order to improve therapeutic efficacy, functionalization of the surface of PtNPs could help to increase biodistribution, accumulation, cell-specific targeting, and controlled release, and reduce side effects to human beings. Though numerous studies highlight the chemotherapeutic effect of platinum-based anticancer drugs, there are relatively few published studies about the radio-sensitizing and synergistic effects of PtNPs for radiation therapy. Hullo et al. [111] showed that PtNPs could induce the radio-enhancement effect in breast cancer cell lines after internalization and accumulate in lysosomes and multivesicular bodies. Likewise, the lysosome-localized PtNPs could absorb radiation energy and focus more on the cancer site, damaging DNA and killing tumor cells. Zhang et al. [112] found that the radiation doses could be physically enhanced when combining the platinum nanoparticles coated with bovine serum albumin (BSA), Pt@BSA NPs, for use in radiotherapy. Also, studies showed that the presence of platinum nanoparticles when cell cultures were irradiated could result in strongly enhanced breaks in DNA, especially DSBs, mediated by water radicals which may originate from the inner-shell excitation of platinum atoms [50,113,114]. In other studies, Gutiérrez et al. [115] discussed the enhanced effect of radiation on cervical–uterine cancer cells (HeLa) when the cancer cells were treated with PEGylated PtNPs functionalized with a fluorescent marker in combination with γ-rays. Results showed that as the radiation dose increased, the number of survived cells decreased in the presence of the nanoparticles. Yang et al. [116] reported that Pt nanoenzyme-functionalized nanoplatform BP/Pt-Ce6@PEG NPs improved the cellular uptake and decomposed endogenous H_2_O_2_ into O_2_ in situ to relieve tumor hypoxia, affording enhanced reactive oxygen species (ROS) production and causing the intratumoral oxygen level to surmount tumor hypoxia for efficient tumor treatment in an in vivo and in vitro study. 

### 2.6. Bismuth-Based Nanoparticles

Meng et al. [117] developed bismuth and gadolinium-codoped carbon quantum dots (Bi, Gd-CQDs) for fluorescence imaging, CT imaging, and MRI imaging. They demonstrated that, due to the high X-ray attenuation coefficient, short T1 relaxation time, and robust and steady fluorescence characteristics of Bi, Gd-CQDs, we could use Bi, Gd-CQDs as a good nanoprobe for CT, MRI, and fluorescence imaging.

It was reported in the literature [118] that triptorelin peptide-targeted multifunctional bismuth nanoparticles (Bi_2_S_3_@BSA-Triptorelin NPs) might be used as a CT contrast agent.

In an investigation into introducing photoacoustic imaging (PAI) contrast agents for deep tissue imaging, Zhao et al. [119] used DNA-templated ultrasmall bismuth sulfide (Bi_2_S_3_) NPs for myocardial infarction imaging. For NPs synthesis, they employed a simple strategy for ultrasmall NPs via self-assembly of single-stranded DNA (ssDNA)/metal ion complexes. Zhao et al. [119] suggested that ultrasmall DNA-Bi_2_S_3_ NPs can be used as a PAI contrast agent for myocardial infarction imaging, and the ssDNA template could be used for ultrasmall PAI contrast agent preparation.

It has now been suggested that polymer-coated bismuth oxychloride (BiOcl) nanosheets can be used as CT contrast agents for gastrointestinal (GI) imaging [120].

In their groundbreaking paper, Zaho et al. [121] developed Bi@mSio_2_@Mno_2_/DOX as a powerful theragnostic agent for CT/MR imaging and photothermal therapy (PPT)/chemodynamic therapy (CDT)/chemotherapy cancer treatment. More recent evidence [122] shows the effect of reducing T_1_ and T_2_ relaxation times and increasing CT image contrast of Bi_2_S_3_@BSA-Fe_3_O_4_ nanoparticle as a dual contrast agent for MRI and CT imaging modalities.

### 2.7. Tantalum-Based Nanoparticles

Lakshmi et al. [123] investigated the impact of tantalum oxide NPs (TaOx NPs) and the Au-decorated tantalum oxide (TaOx-Au NPs) as imaging contrast agents on cancer diagnostics. As Lakshmi et al. [19] noted, TaOx-Au NPs, due to higher X-ray attenuation in a low-energy X-ray, is far more attractive than TaOx NPs and, therefore, can be used for cancer diagnosis with a CT imaging modality. A recent study [124] involved PEG-Ta_2_O_5_@Cus multifunctional NPs for diagnosing hepatocellular carcinoma (HCC) with CT/PA imaging. The application of poly-coated tantalum NPs (Ta@PVP NPs) in medical imaging was first demonstrated by Ji et al. [125]. In their seminal study, Ta@PVP NPs were used as radiotherapy/photothermal therapy (PTT) and CT/PA imaging agents in breast carcinoma.

### 2.8. Ytterbium-Based Nanoparticles

It has now been proposed that [126] glutathione functionalized ytterbium/iron oxide NPs as a dual-modality contrast agent for MRI/CT imaging. In a major advance, Dong et al. [127], for the first time, developed an ultrasmall ytterbium NPs (YbNPs) contrast agent for CT/spectral photon-counting computed tomography (SPECT) imaging. They pointed out that, in the clinical X-ray energy range, the YbNPs attenuation is significantly higher than the AuNPs. Many attempts have been made [128] to introduce multi-modality MRI/PA/NIR-II fluorescence contrast agents. They have focused on using calcium fluoride co-doped with rare-earth ions such as ytterbium, gadolinium, and neodymium (CaF_2_: Yb, Gd, Nd) NPs. It has conclusively been shown that [129] Yb^3+^ concentration in LaNbO_4_ nanoparticles affects the luminescent properties of NPs for medical imaging applications, and the intensity of emissions is directly related to Yb^3+^ concentration. Recently, an in-vivo study has shown that BaYbF_5_-SiO_2_ NPs can be used as contrast agents for imaging the osteochondral interface with micro-CT imaging with high-resolution images [130]. Table 1 presents an overview of some new nanotheranostics and their applications in cancer detection and treatment.

## 3. Conclusions and Future Perspective

This review provides a broad overview of the usage of metal nanoparticles in radiotherapy and medical imaging. As mentioned in this review, early detection and treatment of cancer is a significant public health issue. In medicine, nanoparticles have become important in cancer imaging and treatment for tumor visualization, drug delivery, and direct antitumor potency. In recent years, there has been an increasing interest in using metal nanoparticles in medicine for diagnosis and therapy due to their inherent characteristics such as unique physicochemical properties, high drug payload, size, ability to functionalize easily with biomolecules, electrostatic charge, low toxicity, image contrast enhancement, optical properties, photothermal behavior, high surface area, surface chemistry, and radio-sensitizer properties. Thus, metal NPs are crucial in the medical sciences and play an essential role in cancer detection and treatment. In the current study, the new update of gold-based, gadolinium-based, iron-based, tungsten-based, platinum-based, bismuth-based, tantalum-based, and ytterbium-based NPs was investigated in cancer treatment and detection. Table 1 shows recent primary metal nanoparticles and their application in cancer diagnosis and treatment. Collectively, these studies outline the critical role of metal nanoparticles for cancer treatment and detection with different imaging modalities. Most current studies in metal NPs’ usage for cancer detection and treatment have focused on in vivo and in vitro studies, mainly for determining the physical and chemical characteristics of the synthesized metal NPs. This work attempts to show the recent findings of using metal nanoparticles in medical imaging and therapy. The evidence presented in this study suggests that, based on the type and location of the tumor, a specific nanoparticle can be used for cancer treatment and detection. Taken together, the evidence from this review highlights the role of different parameters, such as NPs’ coating material, size, and synthesized method, for introducing a new and effective contrast agent in cancer imaging or a radio-sensitizer in cancer treatment.

It seems possible that the frequent usage of these metal NPs is due to their high atomic number, easier synthesis, and accessibility. It can therefore be assumed that the high atomic number of metal NPs is responsible for the higher radiation absorption and electron Auger production, which is very important for effective cancer treatment with metal NPs. Thus, they are now the first option for designing and developing metal nanoparticles for medical applications. Although metal NPs have demonstrated high performance and advantages for cancer treatment and detection, they have a particular disadvantage in terms of high aggregation due to high weight, clearance problems, skin color change, and high price for gold-based NPs. While many metal nanoparticles have been discussed in this review, the findings must be interpreted carefully. More research is required to compare and assess these metal nanoparticles’ pharmacokinetics and toxicity properties prior to clinical applications.

Herein, among all metal nanoparticles, gold-based, gadolinium-based, and iron-based nanoparticles are by far the most studied for medical applications such as medical imaging and therapy. It seems possible that these results are due to their ease of functionalization, low toxicity, and superior biocompatibility. Also, iron-based nanoparticles, such as SPIONs, can be used for MPI. MPI is a favorable tool for molecular imaging and detecting SPIONs distribution in tissues.

Unfortunately, discovering and assessing new NPs as candidates for clinical usage in cancer treatment and detection is time-consuming and complicated. In recent years, there has been growing interest in using in-silico studies [131,132,133,134] as a robust process and tool for developing and optimizing proposed NPs in the medical sciences.

This review article could be helpful from an educational point of view for all researchers, in particular medical researchers, who are interested in cancer diagnosis and treatment. The limitation of this article is that it may not cover all recent findings in the world. As seen from the literature review, the toxicity level of metal NPs has been investigated in each study with different tests. These data, the toxicity of NPs, must be interpreted with caution because these studies were conducted in vivo and in vitro and are not clinical studies. Further research should be done to investigate the clinical toxicity of metal NPs in humans. Also, in the author’s view, further research might investigate the safety, biocompatibility, and long-term toxicities of metal NPs in animal models and humans prior to the clinical usage of metal NPs.

## Figures and Tables

**Table 1 diagnostics-13-00833-t001:** Some new nanotheranostics and their applications in medicine.

Authors	Nanoparticles	Application		Conclusion
		Imaging	Therapy	
Tudda et al. [20]	Au NPs		√	AuNPs increase the effective dose in the radiotherapy of breast cancer
Safari et al. [33]	Fe_3_O_4_@Au/Alg NPs		√	improves the photothermal efficiency and dose-enhancement of X-rays
Kitayama et al. [36]	Au MIP-NGs		√	inhibits in-vivo tumor growth with low-dose radiation therapy
Young et al. [42]	Gd-tex^2+^		√	efficient radiation sensitizer in in vitro and in vivo
Zhang et al. [112]	Pt@BSA NPs		√	radiation doses enhancement
Yang et al. [116]	BP/Pt-Ce_6_@PEG NPs		√	radio-sensitization for the hypoxia region of the tumor
Guerra et al. [67]	SPION-DX		√	radio-sensitization for 6 MV photon beam
Chen et al. [85]	Au NPs/UCNPs/WO_3_@C	√	√	theranostics nanoplatform
Mzwd et al. [26]	GA-Au NPs	√	√	laser ablation technique and CT contrast agent
Zhou et al. [17]	Au-UC NPs	√	√	multi-modality imaging and photothermal effect
Zhang et al. [28]	Au-UCNPs-DSPE-PEG_2_k	√	√	MRI and CT contrast agents in vivo and in vitro may also be used for photothermal treatment
Li et al. [27]	Au DENPs labeled with ^68^Ga-DG	√		can be used for PET/CT imaging contrast agent
Baijal et al. [37]	Au-PEG-NPs Ag-PEG-NPs	√	√	Au/Ag-PEG-NPs can be used as a radio-sensitizer and CT contrast agent in oral cancer KB cell lines
Meng et al. [117]	Bi, Gd-CQDs	√		Bi, Gd-CQDs is a good nanoprobe for CT, MRI, and fluorescence imaging
Mohammadi et al. [118]	Bi_2_S_3_@BSA-Triptorelin NPs	√		Bi_2_S_3_@BSA-Triptorelin NPs might be used as a CT contrast agent
Zhao et al. [119]	ultrasmall DNA-Bi_2_S_3_ NPs	√		They suggested that ultrasmall DNA-Bi_2_S_3_ NPs can be used as a PAI contrast agent for myocardial infarction imaging.
Zelepukin et al. [120]	polymer-coated BiOcl nanosheets	√		They suggested that polymer-coated BiOcl nanosheets can be used as CT contrast agents for GI imaging.
Zaho et al. [121]	Bi@mSio_2_@Mno_2_/DOX	√	√	They suggested that Bi@mSio_2_@Mno_2_/DOX is a powerful theragnostic agent for CT/MRI medical imaging and PPT/chemodynamic therapy (CDT)/chemotherapy.
Nosrati et al. [122]	Bi_2_S_3_@BSA-Fe_3_O_4_ NPs	√		They suggested that Bi_2_S_3_@BSA-Fe_3_O_4_ NPs can be used as a dual contrast agent for MRI and CT imaging.

Note: “√” is shows the use and application of each NPs in Imaging, Therapy, or both of them.

## Data Availability

The data presented in this study are available on request from the corresponding author.

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
