# Peer review of "Recent Metal Nanotheranostics for Cancer Diagnosis and Therapy: A Review"

_diagnostics, 2023, doi:10.3390/diagnostics13050833_

Round 1

Reviewer 1 Report

      In this research, the authors reviewed the recent nanotheranostics for cancer diagnosis and therapy. Generally, it’s a meaningful and interesting review but still needs to be improved before possible acception. In my opinion, the current version of this manuscript fits the scope of Diagnostics and could be accepted after major revision.

My specific comments are in detail listed below:

1.     The abstract was poorly written. In my opinion, a better and brief summary and summarize could be more clearly added.

2.     In this review, only Gold, Bismuth, Tungsten, Tantalum, Ytterbium, Gadolinium, Silver, Iron, Platinum, Lead, and In Silico related reviewed. In my opinion, using NIR-I or NIR-II cyanine dyes for cancer diagnosis should be added. Some references should be added, including 10.1002/adma.202206121, doi.org/10.1021/jacs.1c09155, doi.org/10.1016/j.apsb.2022.07.023, and doi.org/10.1021/jacs.0c00734.

3.     All the figures are of low quality. The authors should revise it.

4.     In this review, the review of the recent nanotheranostics for cancer therapy was not enough. More related introduction and review should be added.

5.     Some minor mistakes existed in this paper. The authors should carefully check it.

6.     In this review, the current development of using NIR-I or NIR-II cyanine dyes for cancer therapy (2.9 NIR-I or NIR-II cyanine dyes-based nanoparticles) should be added and more deeply reviewed. Some references should be added, including doi.org/10.1136/gutjnl-2016-311909, doi.org/10.1016/j.cej.2022.140164, doi.org/10.1126/sciadv.abb6104, doi.org/10.1016/j.jconrel.2022.11.004, doi.org/10.1002/adma.201704196, doi.org/10.1186/s12951-021-01124-8, and doi.org/10.1002/adma.201800475.

7.     In the Conclusion and future perspective part, the clinical transformation prospect should be more clearly discussed and revealed.

Author Response

Response to Reviewer #1: 

 Comments and Suggestions for Authors

            In this research, the authors reviewed the recent nanotheranostics for cancer diagnosis and therapy. Generally, it’s a meaningful and interesting review but still needs to be improved before possible acceptation. In my opinion, the current version of this manuscript fits the scope of Diagnostics and could be accepted after major revision.

My specific comments are in detail listed below:

1.The abstract was poorly written. In my opinion, a better and brief summary and summarize could be more clearly added.

At first, I would like to thank you for your valuable comments.

Response: As requested, the abstract was modified in the manuscript.

  1. In this review, only Gold, Bismuth, Tungsten, Tantalum, Ytterbium, Gadolinium, Silver, Iron, Platinum, Lead, and In Silico related reviewed. In my opinion, using NIR-I or NIR-II cyanine dyes for cancer diagnosis should be added. Some references should be added, including 10.1002/adma.202206121, doi.org/10.1021/jacs.1c09155, doi.org/10.1016/j.apsb.2022.07.023, and doi.org/10.1021/jacs.0c00734.

Response: As your recommended, the references were added.

  1. All the figures are of low quality. The authors should revise it.

Response: Images with the original size will be used for article production.

  1. In this review, the review of the recent nanotheranostics for cancer therapy was not enough. More related introduction and review should be added.

Response: As you requested, the new references were added.

  1. Some minor mistakes existed in this paper. The authors should carefully check it.

Response: The manuscript was carefully checked and all changes were added to the manuscript.

  1. In this review, the current development of using NIR-I or NIR-II cyanine dyes for cancer therapy (2.9 NIR-I or NIR-II cyanine dyes-based nanoparticles) should be added and more deeply reviewed. Some references should be added, including doi.org/10.1136/gutjnl-2016-311909, doi.org/10.1016/j.cej.2022.140164, doi.org/10.1126/sciadv.abb6104, doi.org/10.1016/j.jconrel.2022.11.004, doi.org/10.1002/adma.201704196, doi.org/10.1186/s12951-021-01124-8, and doi.org/10.1002/adma.201800475.

Response: We appreciate your valuable comment, as you recommended, the references were added to the manuscript.

  1. In the Conclusion and future perspective part, the clinical transformation prospect should be more clearly discussed and revealed.

Response: Table 1 was transferred to section 2.

Reviewer 2 Report

The current work focuses on a review of the recent nanotheranostics for cancer diagnosis and therapy. The author’s some effort into the manuscript, but major issues should be addressed.

Keywords

The review is based mainly on nanotheranostics, so why not inserted it as Keyword!!

Abstract

- Nanotheranostics based on metal nanoparticles for cancer diagnosis and therapies have been intensively studied. So, first, show the importance of this review and then the main outcomes.

Introduction

- The introduction doesn’t provide sufficient background, and the most relevant references are not included.

- It should discuss the advantages and possible limitations of the metal nanoparticles. Each section of metal nanoparticles shows the advantage and disadvantages of each material as accuracy, stability, expensive, and easier preparation,… 

- Why not mentioned magnetic-particle imaging (MPI) for cancer detection? Compared with existing imaging modalities, magnetic-particle imaging (MPI) is recognized as a favorable tool for cancer diagnosis, as it offers the advantages of zero background signal, zero signal reduction with increasing tissue depth, quantitative linearity, and high sensitivity. Additionally, for MPI, there is no need for ionizing radiation.

Diagnostics. 2021; 11(5):773. https://doi.org/10.3390/diagnostics11050773

- Is any commercial materials approved for clinical? e.g. Resovist and NanoTherm Compare with it for any limitation or disadvantage like accuracy, stability, expensive,…To date, two types of iron oxide NPs have been applied in clinical treatment: (i) shelled with a polysaccharide layer and (ii) shelled with a silica layer.

Nanomaterials 2021, 11, 1096. https://doi.org/10.3390/nano11051096

- The toxicity and availability of Nanotheranostics for biomedical applications are essential and should be clear. A subsection related to toxicity should be inserted to clear this point.

- There is a felt lack of critical assessments by the authors. The authors did not mention the research gap between the previously reported articles and the present situation. In each subsection, authors should incorporate their views to mold the research in a new direction.

- Conclusion and future perspective are very general information. Please rephrase it with the main outputs.

Minor issues

- Line 37, First appearance of abbreviation should have full definition e.g. CT, MRI, radiography, PET, SPECT

- Line 64, where Fig.2. correct numbering of the figure

- In Table 1. Correct typos, and subscripts, of chemical formulas e.g.Fe3O4,..

- References, make one system style for all references

- Ref. No. 44, insert complete and updated the citation

- Ref. No. 45, correct it

- Ref. No. 51, insert complete and update the citation

- Ref. No. 55, insert complete and updated citation

- Ref. No. 60, insert complete and update the citation

- Ref. No. 94, insert complete and updated citation

- Ref. No. 106, insert complete and updated citation

Author Response

Response to Reviewer #2: 

The current work focuses on a review of the recent nanotheranostics for cancer diagnosis and therapy. The author’s some effort into the manuscript, but major issues should be addressed.

Keywords

The review is based mainly on nanotheranostics, so why not inserted it as Keyword!!

Many thanks for your meticulous points. All changes are highlighted with yellow color.

Response: The Keyword was added to the manuscript.

Abstract

- Nanotheranostics based on metal nanoparticles for cancer diagnosis and therapies have been intensively studied. So, first, show the importance of this review and then the main outcomes.

Response: As requested, the abstract was corrected.

Introduction

- The introduction doesn’t provide sufficient background, and the most relevant references are not included.

Response: The references were included in the introduction section.

- It should discuss the advantages and possible limitations of the metal nanoparticles. Each section of metal nanoparticles shows the advantage and disadvantages of each material as accuracy, stability, expensive, and easier preparation,… 

Response: The limitations and advantages of metal NPs were added in the ‘Conclusion and future perspective’ section.

- Why not mentioned magnetic-particle imaging (MPI) for cancer detection? Compared with existing imaging modalities, magnetic-particle imaging (MPI) is recognized as a favorable tool for cancer diagnosis, as it offers the advantages of zero background signal, zero signal reduction with increasing tissue depth, quantitative linearity, and high sensitivity. Additionally, for MPI, there is no need for ionizing radiation.

Diagnostics. 2021; 11(5):773. https://doi.org/10.3390/diagnostics11050773

Response: As requested, the MPI data was added to the manuscript (In 2.3. Iron-based nanoparticles).

- Is any commercial materials approved for clinical? e.g. Resovist and NanoTherm Compare with it for any limitation or disadvantage like accuracy, stability, expensive,…To date, two types of iron oxide NPs have been applied in clinical treatment: (i) shelled with a polysaccharide layer and (ii) shelled with a silica layer.

Nanomaterials 2021, 11, 1096. https://doi.org/10.3390/nano11051096

Response: We appreciate your valuable comment on the studies which included in our study, metal NPs used in In-vivo and In-vitro. However, some of the NPs used in the clinic are included in the manuscript (In 2.3. Iron-based nanoparticles).

- The toxicity and availability of Nanotheranostics for biomedical applications are essential and should be clear. A subsection related to toxicity should be inserted to clear this point.

Response: The requested point was added to the manuscript (‘Conclusion and future perspective’ section in last paragraph).

- There is a felt lack of critical assessments by the authors. The authors did not mention the research gap between the previously reported articles and the present situation. In each subsection, authors should incorporate their views to mold the research in a new direction.

Response: The ‘research gap’ were rewired and highlighted in the manuscript in ‘Introduction’ section. The requested sentences were added to the manuscript (last sentences in the ‘2.1. Gold-based nanoparticles’, ‘2.2. Gadolinium-based nanoparticles’, ‘2.3. Iron-based nanoparticles’ sections.

- Conclusion and future perspective are very general information. Please rephrase it with the main outputs.

Response: Changes were created in the ‘Conclusion and future perspective’ section.

Minor issues

- Line 37, First appearance of abbreviation should have full definition e.g. CT, MRI, radiography, PET, SPECT

- Line 64, where Fig.2. correct numbering of the figure

- In Table 1. Correct typos, and subscripts, of chemical formulas e.g.Fe3O4,..

- References, make one system style for all references

- Ref. No. 44, insert complete and updated the citation

- Ref. No. 45, correct it

- Ref. No. 51, insert complete and update the citation

- Ref. No. 55, insert complete and updated citation

- Ref. No. 60, insert complete and update the citation

- Ref. No. 94, insert complete and updated citation

- Ref. No. 106, insert complete and updated citation

Response: Many thanks for your meticulous points. All minor issues, mentioned above, were corrected in the manuscript.

Round 2

Reviewer 1 Report

The current version of this paper could be accepted.

Reviewer 2 Report

Accept in the present form